# GAUSSIAN PROCESS BEHAVIOUR IN WIDE DEEP NEURAL NETWORKS

**Alexander G. de G. Matthews**
University of Cambridge
am554@cam.ac.uk

**Jiri Hron**
University of Cambridge
jh2084@cam.ac.uk

**Mark Rowland**
University of Cambridge
mr504@cam.ac.uk

**Richard E. Turner**
University of Cambridge
ret26@cam.ac.uk

**Zoubin Ghahramani**
University of Cambridge, Uber AI Labs
zoubin@eng.cam.ac.uk

## ABSTRACT

Whilst deep neural networks have shown great empirical success, there is still much work to be done to understand their theoretical properties. In this paper, we study the relationship between Gaussian processes with a recursive kernel definition and random wide fully connected feedforward networks with more than one hidden layer. We exhibit limiting procedures under which finite deep networks will converge in distribution to the corresponding Gaussian process. To evaluate convergence rates empirically, we use maximum mean discrepancy. We then exhibit situations where existing Bayesian deep networks are close to Gaussian processes in terms of the key quantities of interest. Any Gaussian process has a flat representation. Since this behaviour may be undesirable in certain situations we discuss ways in which it might be prevented. [1]

## 1    INTRODUCTION

Deep feedforward neural networks have emerged as an essential component of modern machine learning. As such there has been significant research effort in trying to understand the theoretical properties of such models. One important branch of such research is the study of random networks. By assuming a probability distribution on the network parameters, a distribution is induced on the input to output function that such networks encode. This has proved important in the study of initialisation and learning dynamics (Schoenholz et al., 2017) and expressivity (Poole et al., 2016). It is, of course, essential in the study of Bayesian priors on networks (Neal, 1996). The Bayesian approach makes little sense if prior assumptions are not understood, and distributional knowledge can be essential in finding good posterior approximations.

Since we typically want our networks to have high modelling capacity, it is natural to consider limit distributions of networks as they become large. Whilst distributions on deep networks are generally challenging to work with exactly, the limiting behaviour can lead to more insight. Further, as we shall see, networks used in the literature may be very close to this behaviour.

The seminal work in this area is that of Neal (1996), which showed that under certain conditions random neural networks with one hidden layer converge to a Gaussian process. The question of the type of convergence is non-trivial and part of our discussion. Historically this result was a significant one because it provided a connection between flexible Bayesian neural networks and Gaussian processes (Williams, 1998; Rasmussen & Williams, 2006)

---

[1] Code for the experiments in the paper can be found at https://github.com/widedeepnetworks/widedeepnetworks

## 1.1 OUR CONTRIBUTIONS

We extend the theoretical understanding of random fully connected networks and their relationship to Gaussian processes. In particular, we prove a rigorous result (Theorem 1) on the convergence of certain finite networks with more than one hidden layer to Gaussian processes.

Further, we empirically study the distance between finite networks and their Gaussian process analogues by using maximum mean discrepancy (Gretton et al., 2012) as a distance measure. We find that Bayesian deep networks from the literature can exhibit predictions that are close to Gaussian processes. To demonstrate this, we systematically compare exact Gaussian process inference with 'gold standard' MCMC inference for Bayesian neural networks.

Our work is of relevance to the theoretical understanding of neural network initialisation and dynamics. It is also important in the area of Bayesian deep networks because it demonstrates that Gaussian process behaviour can arise in more situations of practical interest than previously thought. If this behaviour is desired then Gaussian process inference (exact and approximate) should also be considered. In some scenarios, the behaviour may not be desired because it implies a lack of a hierarchical representation. We therefore highlight promising ideas from the literature to prevent such behaviour.

## 1.2 RELATED WORK

The case of random neural networks with one hidden layer was studied by Neal (1996). Cho & Saul (2009) provided analytic expressions for single layer kernels including those corresponding to a rectified linear unit (ReLU). They also studied recursive kernels designed to 'mimic computation in large, multilayer neural nets'. As discussed in Section 3 they arrived at the correct kernel recursion through an erroneous argument. Such recursive kernels were later used with empirical success in the Gaussian process literature (Krauth et al., 2017), with a similar justification to that of Cho and Saul. The first case we are aware of using a Gaussian process construction with more than one hidden layer is the work of Hazan & Jaakkola (2015). Their contribution is similar in content to Lemma 1 discussed here, and the work has had increasing interest from the kernel community (Mitrovic et al., 2017). Recent work from Daniely et al. (2016) uses the concept of 'computational skeletons' to give concentration bounds on the difference in the second order moments of large finite networks and their kernel analogue, with strong assumptions on the inputs. The Gaussian process view given here, without strong input assumptions, is related but concerns not just the first two moments of a random network but the full distribution. As such the theorems we obtain are distinct. A less obvious connection is to the recent series of papers studying deep networks using a mean field approximation (Poole et al., 2016; Schoenholz et al., 2017). In those papers a second order approximation gives equivalent behaviour to the kernel recursion. By contrast, in this paper the claim is that the behaviour emerges as a consequence of increasing width and is therefore something that needs to be proved. Another surprising connection is to the analysis of self-normalizing neural networks (Klambauer et al., 2017). In their analysis the authors assume that the hidden layers are wide in order to invoke the central limit theorem. The premise of the central limit theorem will only hold approximately in layers after the first one and this theoretical barrier is something we discuss here. An area that is less related than might be expected is that of 'Deep Gaussian Processes' (DGPs) (Damianou & Lawrence, 2013). As will be discussed in Section 6, narrow intermediate representations mean that the marginal behaviour is not close to that of a Gaussian process. Duvenaud et al. (2014) offer an analysis that largely applies to DGPs though they also study the Cho and Saul recursion with the motivating argument from the original paper.

## 2 THE DEEP WIDE LIMIT

We consider a fully connected network as shown in Figure 1. The inputs and outputs will be real valued vectors of dimension $M$ and $L$ respectively. The network is fully connected. The initial step and recursion are standard. The initial step is:

$$f_i^{(1)}(x) = \sum_{j=1}^{M} w_{i,j}^{(1)} x_j + b_i^{(1)} . \tag{1}$$

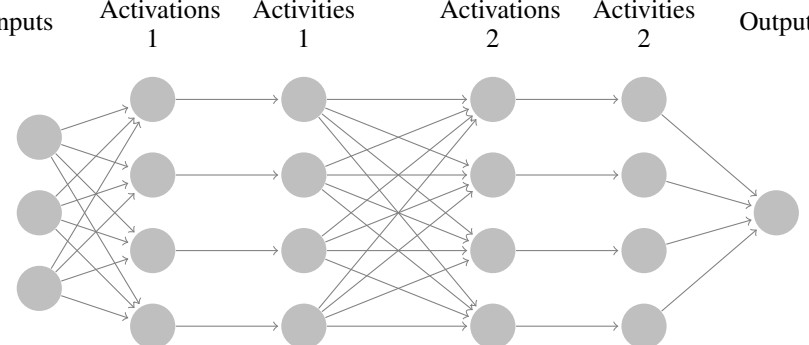

Figure 1: In this paper we consider fully connected feedforward networks with more than one hidden layer. We call the pre-nonlinearity an *activation* and post-nonlinearity an *activity*. As the network becomes increasingly wide the distribution of the marginal distributions of the activations at each layer and of the output will become close to a Gaussian process in a sense described in the text.

We make the functional dependence on $x$ explicit in our notation as it will help clarify what follows. For a network with $D$ hidden layers the recursion is, for each $\mu = 1, \ldots, D$,

$$g_i^{(\mu)}(x) = \phi(f_i^{(\mu)}(x)) \tag{2}$$

$$f_i^{(\mu+1)}(x) = \sum_{j=1}^{H_\mu} w_{i,j}^{(\mu+1)} g_j^{(\mu)}(x) + b_i^{(\mu+1)}, \tag{3}$$

so that $f^{(D+1)}(x)$ is the output of the network given input $x$. $\phi$ denotes the non-linearity. In all cases the equations hold for each value of $i$; $i$ ranges between 1 and $H_\mu$ in Equation (2), and between 1 and $H_{\mu+1}$ in Equation (3) except in the case of the final activation where the top value is $L$. The network could of course be modified to be probability simplex-valued by adding a softmax at the end.

A distribution on the parameters of the network will be assumed. Conditional on the inputs, this induces a distribution on the activations and activities. In particular we will assume independent normal distributions on the weights and biases

$$w_{i,j}^{(\mu)} \sim \mathcal{N}(0, C_w^{(\mu)}) \text{ indep} \tag{4}$$

$$b_i^{(\mu)} \sim \mathcal{N}(0, C_b^{(\mu)}) \text{ indep.} \tag{5}$$

We will be interested in the behaviour of this network as the widths $H_\mu$ becomes large. The weight variances for $\mu \geq 2$ will be scaled according to the width of the network to avoid a divergence in the variance of the activities in this limit. As will become apparent, the appropriate scaling is

$$C_w^{(\mu)} = \frac{\hat{C}_w^{(\mu)}}{H_\mu} \quad \mu \geq 2. \tag{6}$$

The assumption is that $\hat{C}_w^{(\mu)}$ will remain fixed as we take the limit. Neal (1996) analysed this problem for $D = 1$, showing that as $H_1 \to \infty$, the values of $f_i^{(2)}(x)$, the output of the network in this case, converge to a certain multi-output Gaussian process if the activities have bounded variance.

Since our approach relies on the multivariate central limit theorem we will arrange the relevant terms into (column) vectors to make the linear algebra clearer. Consider any two inputs $x$ and $x'$ and all

output functions ranging over the index $i$. We define the vector $f^{(2)}(x)$ of length $L$ whose elements are the numbers $f_i^{(2)}(x)$. We define $f^{(2)}(x')$ similarly. For the weight matrices defined by $w_{i,j,}^{(\mu)}$ for fixed $\mu$ we use a 'placeholder' index $\cdot$ to return column and row vectors from the weight matrices. In particular $w_{j,\cdot}^{(1)}$ denotes row $j$ of the weight matrix at depth 1. Similarly, $w_{\cdot,j}^{(2)}$ denotes column $j$ at depth 2. The biases are given as column vectors $b^{(1)}$ and $b^{(2)}$. Finally we concatenate the two vectors $f^{(2)}(x)$ and $f^{(2)}(x')$ into a single column vector $F^{(2)}$ of size $2L$. The vector in question takes the form

$$F^{(2)} = \begin{pmatrix} f^{(2)}(x) \\ f^{(2)}(x') \end{pmatrix} = \begin{pmatrix} b^{(2)} \\ b^{(2)} \end{pmatrix} + \sum_{j=1}^{H_1} \begin{pmatrix} w_{\cdot,j}^{(2)} \phi(w_{j,\cdot}^{(1)} x + b_j^{(1)}) \\ w_{\cdot,j}^{(2)} \phi(w_{j,\cdot}^{(1)} x' + b_j^{(1)}) \end{pmatrix} \tag{7}$$

The benefit of writing the relation in this form is that the applicability of the multivariate central limit theorem is immediately apparent. Each of the vector terms on this right hand side is independent and identically distributed conditional on the inputs $x$ and $x'$. By assumption, the activities have bounded variance. The scaling we have chosen on the variances is precisely that required to ensure the applicability of the theorem. Therefore as $H$ becomes large $F^{(2)}$ converges in distribution to a multivariate normal distribution. The limiting normal distribution is fully specified by its first two moments. Defining $\gamma \sim \mathcal{N}(0, C_b^{(1)}), \epsilon \sim \mathcal{N}(0, C_w^{(1)} I_M)$, the moments in question are:

$$\mathbb{E}\left[ f_i^{(2)}(x) \right] = 0 \tag{8}$$

$$\mathbb{E}\left[ f_i^{(2)}(x) f_j^{(2)}(x') \right] = \delta_{i,j} \left[ \hat{C}_w^{(2)} \mathbb{E}_{\epsilon,\gamma} \left[ \phi(\epsilon^T x + \gamma) \phi(\epsilon^T x' + \gamma) \right] + C_b^{(2)} \right] \tag{9}$$

Note that we could have taken a larger set of input points to give a larger vector $F$ and again we would conclude that this vector converged in distribution to a multivariate normal distribution. More formally, we can consider the set of possible inputs as an *index set*. A set of consistent finite dimensional Gaussian distributions on an index set corresponds to a Gaussian process by the Kolmogorov extension theorem. The Gaussian process in question is a distribution over functions defined on the product $\sigma$-algebra, which has the relevant finite dimensional distributions as its marginals.

In the case of a multivariate normal distribution a set of variables having a covariance of zero implies that the variables are mutually independent. Looking at Equation (9), we see that the limiting distribution has independence between different components $i, j$ of the output. Combining this with the recursion (2), we might intuitively suggest that the next layer also converges to a multivariate normal distribution in the limit of large $H_\mu$. Indeed we state the following lemma, which we attribute to Hazan & Jaakkola (2015):

**Lemma 1** (Normal recursion)**.** *If the activations of a previous layer are normally distributed with moments:*

$$\mathbb{E}\left[ f_i^{(\mu-1)}(x) \right] = 0 \tag{10}$$

$$\mathbb{E}\left[ f_i^{(\mu-1)}(x) f_j^{(\mu-1)}(x') \right] = \delta_{i,j} K(x, x'), \tag{11}$$

*Then under the recursion (2) and as $H \to \infty$ the activations of the next layer converge in distribution to a normal distribution with moments*

$$\mathbb{E}\left[ f_i^{(\mu)}(x) \right] = 0 \tag{12}$$

$$\mathbb{E}\left[ f_i^{(\mu)}(x) f_j^{(\mu)}(x') \right] = \delta_{i,j} \left[ \hat{C}_w^{(\mu)} \mathbb{E}_{(\epsilon_1,\epsilon_2) \sim \mathcal{N}(0,K)} [\phi(\epsilon_1)\phi(\epsilon_2)] + C_b^{(\mu)} \right] \tag{13}$$

*where $K$ is a $2 \times 2$ matrix containing the input covariances.*

Unfortunately the lemma is not sufficient to show that the joint distribution of the activations of higher layers converge in distribution to a multivariate normals. This is because for finite $H$ the input activations do not have a multivariate normal distribution - this is only attained (weakly or in distribution) in the limit. It could be the case that the *rate* at which the limit distribution is attained affects the distribution in subsequent layers. We are able to offer the following theorem rigorously:

**Theorem 1.** *Consider a Bayesian deep neural network of the form in Equations* (1) *and* (2) *using ReLU activation functions. Then there exist strictly increasing width functions* $h_\mu : \mathbb{N} \mapsto \mathbb{N}$ *such that* $H_1 = h_1(n), \ldots, H_D = h_D(n)$, *and for any countable input set* $(x[i])_{i=1}^\infty$, *the distribution of the output of the network converges in distribution to a Gaussian process as* $n \to \infty$.

A proof is included in the appendix. We conjecture that a more general theorem will hold. In particular we expect that the width functions $h_\mu$ can be taken to be the identity and that the non-linearity can be extended to monotone functions with well behaved tails. Our conjecture is based on the intuition from Lemma 1 and from our experiments, in which we always take the width function to be the identity.

## 3 SPECIFIC KERNELS UNDER RECURSION

Cho & Saul (2009) suggest a family of kernels based on a recurrence designed to 'mimic computation in large, multilayer neural nets'. It is therefore of interest to see how this relates to deep wide Gaussian processes. A kernel may be associated with a feature mapping $\Phi(x)$ such that $K(x, x') = \Phi(x) \cdot \Phi(x')$. Cho and Saul define a recursive kernel through a new feature mapping by compositions such as $\Phi(\Phi(x))$. However this cannot be a legitimate way to create a kernel because such a composition represents a type error. There is no reason to think the output dimension of the function $\Phi$ matches the input dimension and indeed the output dimension may well be infinite. Nevertheless, the paper provides an elegant solution to a different task: it derives closed form solution to the recursion from Lemma 1 (Hazan & Jaakkola, 2015) for the special case

$$\phi(u) = \Theta(u)u^r \ \text{ for } \ r = 0, 1, 2, 3, \tag{14}$$

where $\Theta$ is the Heaviside step function. Specifically, the recursive approach of Cho & Saul (2009) can be adapted by using the fact that $u^\top z$ for $z \sim \mathcal{N}(0, LL^\top)$ is equivalent in distribution to $(L^\top u)^\top \varepsilon$ with $\varepsilon \sim \mathcal{N}(0, I)$, and by optionally augmenting $u$ to incorporate the bias. Since $r = 1$ corresponds to rectified linear units we apply this analytic kernel recursion in all of our experiments.

## 4 MEASURING CONVERGENCE USING MAXIMUM MEAN DISCREPANCY

In this section we use the kernel based two sample tests of Gretton et al. (2012) to empirically measure the similarity of finite random neural networks to their Gaussian process analogues. The maximum mean discrepancy (MMD) between two distributions $\mathcal{P}$ and $\mathcal{Q}$ is defined as:

$$\mathcal{MMD}(\mathcal{P}, \mathcal{Q}, \mathcal{H}) := \sup_{||h||_{\mathcal{H}} \leq 1} \left[ \mathbb{E}_{\mathcal{P}}[h] - \mathbb{E}_{\mathcal{Q}}[h] \right] \tag{15}$$

where $\mathcal{H}$ denotes a reproducing kernel Hilbert space and $|| \cdot ||_{\mathcal{H}}$ denotes the corresponding norm. It gives the biggest possible difference between expectations of a function under the two distributions under the constraint that the function has Hilbert space norm less than or equal to one. We used the unbiased estimator of squared MMD given in Equation (3) of Gretton et al. (2012).

In this experiment and all those that follow we take weight variance parameters $\hat{C}_w^{(\mu)} = 0.8$ and bias variance $C_b = 0.2$. We took 10 standard normal input points in 4 dimensions and pass them through 2000 independent random neural networks drawn from the distribution discussed in this paper. This was then compared to 2000 samples drawn from the corresponding Gaussian process distribution. The experiment was performed with different numbers of hidden layers and numbers of units per hidden layer. We repeated each experiment 20 times which allows us to reduce variance in our results and give a simple estimate of measurement error. The experiments use an RBF kernel for the MMD

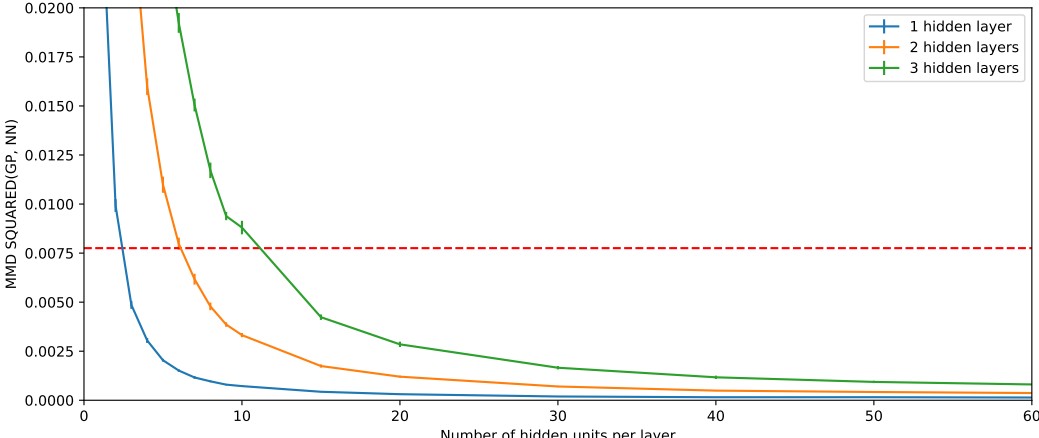

Figure 2: A comparison of finite random neural networks to their corresponding Gaussian process analogue using an (RBF) kernel estimator of the squared maximum mean discrepancy (MMD). The results are consistent with the emergence of Gaussian process behaviour as the networks become wide. The red dashed line is for calibration and denotes the squared MMD between two Gaussian processes with isotropic RBF kernels and length scales $l$ and $2l$ where $l = \sqrt{8}$ is the characteristic length scale of the input space (see text).

estimate with lengthscale $1/2$. In order to help give an intuitive sense of the distances involved we also include a comparison between two Gaussian processes with isotropic RBF kernels using the same MMD distance measure. The kernel length scales for this pair of 'calibration' Gaussian processes are taken to be $l$ and $2l$, where the characteristic length scale $l = \sqrt{8}$ is chosen to be sensible for the standard Normal input distribution on the four dimensional space.

The results of the experiment are shown in Figure 2. We see that for each fixed depth the network converges towards the corresponding Gaussian process as the width increases. For the same number of hidden units per layer, the MMD distance between the networks and their Gaussian process analogue becomes higher as depth increases. The rate of convergence to the Gaussian process is slower as the number of hidden layers is increased.

## 5    COMPARING BAYESIAN DEEP NETWORKS TO GAUSSIAN PROCESSES

In this section we compare the behaviour of finite Bayesian deep networks of the form considered in this paper with their Gaussian process analogues. If we make the networks wide enough the agreement will be very close. It is also of interest, however, to consider the behaviour of networks actually used in the literature, so we use 3 hidden layers and 50 hidden units which is typical of the networks used by Hernández-Lobato & Adams (2015). Fully connected Bayesian deep networks with finite variance priors on the weights have also been considered in other works (Graves, 2011; Hernández-Lobato et al., 2016; Blundell et al., 2015), though the specific details vary. We use rectified linear units and correct the variances to avoid a loss of prior variance as depth is increased. Our general strategy was to compare exact Gaussian process inference against expensive 'gold standard' Markov Chain Monte Carlo (MCMC) methods. We choose the latter because used correctly it works well enough to largely remove questions of posterior approximation quality from the calculus of comparison. It does mean however that our empirical study does not extend to datasets which are large in terms of number of data points or dimensionality, where such inference is challenging. We therefore sound a note of caution about extrapolating our empirical finite network conclusions too confidently to this domain. On the other hand, lower dimensional, prior dominated problems are generally regarded as an area of strength for Bayesian approaches and in this context our results are directly relevant.

We computed the posterior moments by the two different methods on some example datasets. For the MCMC we used Hamiltonian Monte Carlo (HMC) (Neal, 2010) updates interleaved with elliptical

slice sampling (Murray et al., 2010). We considered a simple one dimensional problem and a two dimensional real valued embedding of the four data point XOR problem. We see in Figures 3 and 4 (left) that the agreement in the posterior moments between the Gaussian process and the Bayesian deep network is very close.

A key quantity of interest in Bayesian machine learning is the marginal likelihood. It is the normalising constant of the posterior distribution and gives a measure of the model fit to the data. For a Bayesian neural network, it is generally very difficult to compute, but with care and computational time it can be approximated using Hamiltonian annealed importance sampling (Sohl-Dickstein & Culpepper, 2012). The log-importance weights attained in this way constitute a stochastic lower bound on the marginal likelihood (Grosse et al., 2015). Figure 4 (right) shows the result of such an experiment compared against the (extremely cheap) Gaussian process marginal likelihood computation on the XOR problem. The value of the log-marginal likelihood computed in the two different ways agree to within a single nat which is negligible from a model selection perspective (Grosse et al., 2015).

Predictive log-likelihood is a measure of the quality of probabilistic predictions given by a Bayesian regression method on a test point. To compare the two models we sampled $10$ standard normal train and test points in $4$ dimensions and passed them through a random network of the type under study to get regression targets. We then discarded the true network parameters and compared the predictions of posterior inference between the two methods. We also compared the marginal predictive distributions of a latent function value. Figure 5 shows the results. We see that the correspondence in predictive log-likelihood is close but not exact. Similarly the marginal function values are close to those of a Gaussian process but are slightly more concentrated.

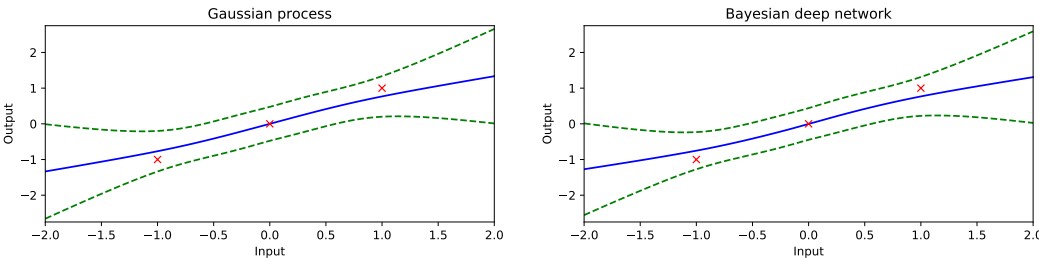

Figure 3: A comparison between Bayesian posterior inference in a Bayesian deep neural network and posterior inference in the analogous Gaussian process. The neural network has 3 hidden layers and $50$ units per layer. The lines show the posterior mean and two $\sigma$ credible intervals.

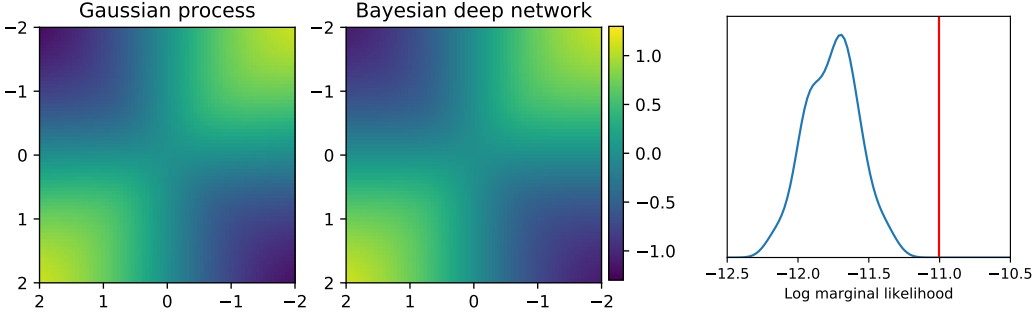

Figure 4: A comparison between posterior inference for a Gaussian process and a Bayesian deep network for a real value embedding of the XOR function. Left and centre: The two posterior means. The mean absolute different between the two posterior estimate grids is $0.027$. Right: Kernel density estimate of the log weights from annealed importance sampling on a Bayesian deep network compared to the analogous Gaussian process marginal likelihood shown by the vertical line. The neural network has 3 hidden layers and $50$ units per layer.

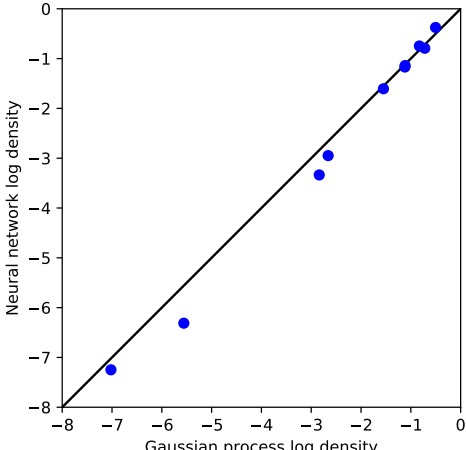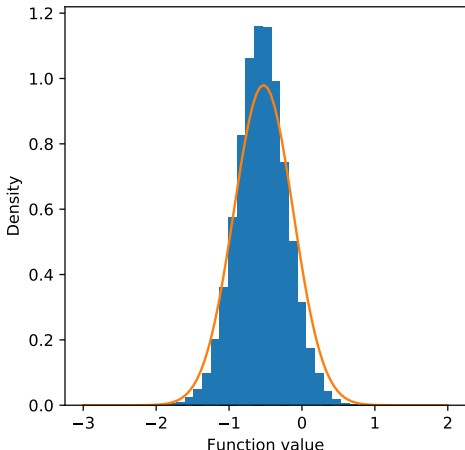

Figure 5: A comparison of the predictive distributions of a Bayesian deep network and a Gaussian process on a randomly generated test case. Left: the per point log-densities of the two models. Right: a randomly selected predictive marginal distribution for the latent function on a randomly selected test point.

## 6 AVOIDING GAUSSIAN PROCESS BEHAVIOUR

When using deep Bayesian neural networks as priors, the emergence of Gaussian priors raises important questions in the cases where it is applicable, even if one sets aside questions of computational tractability. It has been argued in the literature that there are important cases where kernel machines with local kernels will perform badly (Bengio et al., 2005). The analysis applies to the posterior mean of a Gaussian process. The emergent kernels in our case are hyperparameter free. Although they do not meet the strict definition of what could be considered 'local' the fact remains that any Gaussian process with a fixed kernel does not use a learnt hierarchical representation. Such representations are widely regarded to be essential to the success of deep learning. There is relevant literature here on learning the representation of a standard, usually structured, network composed with a Gaussian process (Wilson et al., 2016a;b; Al-Shedivat et al., 2017). This differs from the assumed paradigm of this paper, where all model complexity is specified probabilistically and we do not assume convolutional, recurrent or other problem specific structure.

Within this paradigm, the question therefore arises as to what can be done to avoid marginal Gaussian process behaviour if it is not desired. Speaking loosely, to stop the onset of the central limit theorem and the approximate analogues discussed in this paper one needs to make sure that one or more of its conditions is far from being met. Since the chief conditions on the summands are independence, bounded variance and many terms, violating these assumptions will remove Gaussian process behaviour. Deep Gaussian processes (Damianou & Lawrence, 2013) are not close to standard Gaussian processes marginally because they are typically used with narrow intermediate layers. It can be challenging to choose the precise nature of these narrow layers a priori. Neal (1996) suggests using networks with infinite variance in the activities. With a single hidden layer and correctly scaled, these networks become alpha stable processes in the wide limit. Neal also discusses variants that destroy independence by coupling weights. Our results about the emergence of Gaussian processes even with more than one hidden layer mean these ideas are of considerable interest going forward.

## 7 CONCLUSIONS

Studying the limiting behaviour of distributions on feedforward networks has been a fruitful avenue for understanding these models historically. In this paper we have extended the state of knowledge about the wide limit, including for networks with more than one hidden layer. In particular, we have exhibited limit sequences of networks that converge in distribution to Gaussian processes with a certain recursively defined kernel. Our empirical study using MMD suggests that this behaviour

is exhibited in a variety of models of size comparable to networks used in the literature. This led us to juxtapose finite Bayesian neural networks with their Gaussian process analogues, finding that the agreement in terms of key predictors is close empirically. If this Gaussian process behaviour is desired then exact and approximate inference using the analytic properties of Gaussian processes should be considered as an alternative to neural network inference. Since Gaussian processes have an equivalent flat representation then in the context of deep learning the behaviour may well not be desired and steps should be taken to avoid it.

We view these results as a new opportunity to further the understanding of neural networks in the work that follows. Initialisation and learning dynamics are crucial topics of study in modern deep learning which require that we understand random networks. Bayesian neural networks should offer a principled approach to generalisation but this relies on successfully approximating a clearly understood prior. In illustrating the continued importance of Gaussian processes as limit distributions, we hope that our results will further research in these broader areas.

## 8 ACKNOWLEDGEMENTS

We wish to thank Neil Lawrence for helpful conversations. We also thank the anonymous reviewers for their insights. Alexander Matthews and Zoubin Ghahramani acknowledge the support of EPSRC Grant EP/N014162/1 and EPSRC Grant EP/N510129/1 (The Alan Turing Institute). Jiri Hron holds a Nokia CASE Studentship. Mark Rowland acknowledges support by EPSRC grant EP/L016516/1 for the Cambridge Centre for Analysis. Richard E. Turner is supported by Google as well as EPSRC grants EP/M0269571 and EP/L000776/1.

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

# A PROOF OF MAIN THEOREM

## A.1 STATEMENT OF THEOREM AND NOTATION

In this section, we provide a proof of the main theorem of the paper, which we begin by recalling.

**Theorem 1.** *Consider a Bayesian deep neural network of the form in Equations* (1) *and* (2) *using ReLU activation functions. Then there exist strictly increasing width functions* $h_\mu : \mathbb{N} \mapsto \mathbb{N}$ *such that* $H_1 = h_1(n), \ldots, H_D = h_D(n)$, *and for any countable input set* $(x[i])_{i=1}^\infty$, *the distribution of the output of the network converges in distribution to a Gaussian process as* $n \to \infty$.

The theorem is proven via use of the propositions that follow below. The broad structure of the proof is to use a particular variant of the Berry-Esseen inequality to upper bound how far each layer is from a multivariate normal distribution, and then to inductively propagate these inequalities through the network, leading to a bound on the distance between the output of the network for a collection of input points, and a multivariate Gaussian distribution. These notions will be made precise below. We begin in Section A.2 by stating the propositions that will be used in the proof of Theorem 1, but first establish notation that will be used in the remainder of the appendix.

Given a finite set of inputs $x[1], \ldots, x[n] \in \mathbb{R}^M$, we will write:

- $f^{(\mu)}(\mathbf{x})$ for the random variables $(f^{(\mu)}(x[i]))_{i=1}^n$ collectively taking values in $\mathbb{R}^{nH_\mu}$;
- $f_j^{(\mu)}(\mathbf{x})$ for the random variables $(f_j^{(\mu)}(x[i]))_{i=1}^n$ collectively taking values in $\mathbb{R}^n$;
- $g^{(\mu)}(\mathbf{x})$ for the random variables $(g^{(\mu)}(x[i]))_{i=1}^n$ collectively taking values in $\mathbb{R}^{nH_\mu}$;
- $g_j^{(\mu)}(\mathbf{x})$ for the random variables $(g_j^{(\mu)}(x[i]))_{i=1}^n$ collectively taking values in $\mathbb{R}^n$;

Throughout, if $U_1, U_2$ are random variables taking in values in some Euclidean space $\mathbb{R}^d$, we will define

$$d(U_1, U_2) = \sup_{\substack{A \subseteq \mathbb{R}^d \\ A \text{ convex}}} |\mathbb{P}(U_1 \in A) - \mathbb{P}(U_2 \in A)| .$$

Note that convergence of a sequence of random variables in this metric implies convergence in distribution.

We will also consider multivariate normal distributions $(Z_j^{(\mu)}(x[i]) | j = 1, \ldots, H_\mu, \ i = 1, \ldots, n)$ with covariance matrices of block diagonal form, such that

$$\text{Cov}(Z_k^{(\mu)}(x[a]), Z_l^{(\mu)}(x[b])) = 0 \text{ for distinct } k, l \in \{1, \ldots, H_\mu\}, \quad \text{for all } x[a], x[b] .$$

To avoid writing this in full every time it is required, we will refer to this condition as blockwise independence with respect to the index $j$. We will avoid specification of all covariance values, deferring to the expression (13) given in the main paper. Finally, to simplify notation, we will assume that the network output is one-dimensional. Our proof trivially extends to arbitrary finite output dimension where the limiting distribution is a coordinate-wise independent multivariate GP.

## A.2 SUPPORTING RESULTS

**Proposition 1.** *Let* $\varepsilon > 0$, *and* $x[1], \ldots, x[n] \in \mathbb{R}^M$. *Let* $\mu \in \{2, \ldots, D+1\}$, *and let* $H_k = 2^{H_{k+1}^2}$ *for* $k = 1, \ldots, D-1$. *Then for* $H_D$ *sufficiently large, suppose the condition*

$$d(f^{(\mu-1)}(\mathbf{x}), Z^{(\mu-1)}(\mathbf{x})) \le 2^{-((D+1)-(\mu-1))-n \sum_{k=\mu-1}^D H_k} \varepsilon ,$$

*holds, where* $Z^{(\mu-1)}(\mathbf{x}) = (Z_j^{(\mu-1)}(x[i]) | j = 1, \ldots, H_{\mu-1}, \ i = 1, \ldots, n)$ *is mean-zero multivariate normal, with blockwise independence with respect to the index* $j$. *Then we have*

$$d(f^{(\mu)}(\mathbf{x}), Z^{(\mu)}(\mathbf{x})) \le 2^{-((D+1)-\mu)-n \sum_{k=\mu}^D H_k} \varepsilon ,$$

*where* $Z^{(\mu)}(\mathbf{x}) = (Z_j^{(\mu)}(x[i]) | j = 1, \ldots, H_\mu, \ i = 1, \ldots, n)$ *is mean-zero multivariate normal, with blockwise independence with respect to the index* $j$.

**Proposition 2.** *Let $\varepsilon > 0$, and $x[1], \ldots, x[n] \in \mathbb{R}^M$. If $H_k = 2^{H_{k+1}^2}$ for $k = 1, \ldots, D-1$, then for $H_D$ sufficiently large, we have*

$$d(f^{(D+1)}(\mathbf{x}), Z(\mathbf{x})) \leq \varepsilon,$$

*where $Z(\mathbf{x})$ is a mean-zero multivariate normal random variable.*

In establishing the two propositions above, the following three lemmas will be useful.

**Lemma 2.** *Let $\varepsilon > 0$, and let $Z^{(\mu-1)}(\mathbf{x}) = (Z_j^{(\mu-1)}(x[i]) | j = 1, \ldots, H_{\mu-1}, \ i = 1, \ldots, n)$ be mean-zero multivariate normal, with blockwise independence with respect to the index $j$. Let $\widetilde{g}^{(\mu-1)}(\mathbf{x}) = \phi(Z^{(\mu-1)}(\mathbf{x}))$, and let $\widetilde{f}^{(\mu)}(\mathbf{x})$ be given by*

$$\widetilde{f}^{(\mu)}(x[i]) = \sum_{j=1}^{H_{\mu-1}} w_{\cdot,j}^{(\mu)} \widetilde{g}_j^{(\mu-1)}(x[i]) + b^{(\mu)},$$

*for $i = 1, \ldots, n$. Then given $\varepsilon > 0$, if $H_k = 2^{H_{k+1}^2}$ for $k = 1, \ldots, D-1$, then for all sufficiently large $H_D$ we have:*

$$d(\widetilde{f}^{(\mu)}(\mathbf{x}), Z^{(\mu)}(\mathbf{x})) \leq 2^{-((D+1)-(\mu-1))-n\sum_{k=\mu}^{D} H_k}\varepsilon,$$

*where $Z^{(\mu)}(\mathbf{x}) = (Z_j^{(\mu)}(x[i]) | j = 1, \ldots, H_\mu, \ i = 1, \ldots, n)$ is mean-zero multivariate normal, with blockwise independence with respect to the index $j$.*

**Lemma 3.** *Let $Z^{(\mu-1)}(\mathbf{x}) = (Z_j^{(\mu-1)}(x[i]) | j = 1, \ldots, H_{\mu-1}, 1, \ldots, n)$ be mean-zero multivariate normal, with blockwise independence with respect to the index $j$, such that for some $\varepsilon > 0$,*

$$d(Z^{(\mu-1)}(\mathbf{x}), f^{(\mu-1)}(\mathbf{x})) \leq \varepsilon.$$

*Then, defining $\widetilde{f}^{(\mu)}(\mathbf{x})$ by*

$$\widetilde{f}^{(\mu)}(x[i]) = \sum_{j=1}^{H_\mu} w_{\cdot,j} \phi(Z_j^{(\mu-1)}(x[i])) + b^{(\mu)},$$

*in the particular case where $\phi$ is the elementwise ReLU function, we have*

$$d(\widetilde{f}^{(\mu)}(\mathbf{x}), f^{(\mu)}(\mathbf{x})) \leq 2^{nH_{\mu-1}}\varepsilon.$$

**Lemma 4.** *Let $X_1, \ldots, X_{H_{\mu-1}}$ be iid random variables of the form $X_j = \widetilde{g}_j^{(\mu-1)}(\mathbf{x}) \otimes \widetilde{w}_{\cdot,j}^{(\mu)}$, where $\otimes$ denotes the Kronecker product, $\widetilde{g}_j^{(\mu-1)}(\mathbf{x})$ is defined as in Lemma 2, and $\widetilde{w}_{\cdot,j}^{(\mu)}$ is a multivariate normal variable taking values in $\mathbb{R}^{H_\mu}$ with mean vector 0, and covariance $\hat{C}_w^{(\mu)} I$. We denote the variance of $X_j$ by $\Sigma_\otimes$ and its Schur decomposition as $\Sigma_\otimes = Q_\otimes \Lambda_\otimes Q_\otimes^T$. Then $\beta = \mathbb{E}\left[\|Q_\otimes \Lambda_\otimes^{-1/2} Q_\otimes^T X_j\|^3\right] \leq C_{H_\mu, n}$, where $C_{H_\mu, n} \in \mathbb{R}$ depends on $H_\mu$ and $n$, but is independent of $H_{\mu-1}$. Further, we have $C_{H_\mu, n} = \mathcal{O}(H_\mu^2 n^2)$.*

### A.3 PROOFS

*Proof of Lemma 2.* We use a straightforward variant of a particular Berry-Esseen inequality described in Bentkus (2003). We first state this result from the literature, and then derive a straightforward variation that we will use in the sequel.

**Theorem 2** (From Bentkus (2003))**.** *Let $X_1, \ldots, X_n$ be iid random variables taking values in $\mathbb{R}^d$, with mean vector 0, identity covariance matrix, and $\beta = \mathbb{E}\left[\|X_i\|^3\right] < \infty$. Let $S_n = \frac{1}{\sqrt{n}}\sum_{i=1}^n X_i$, and let $Y$ be a standard $d$-dimensional multivariate normal random vector. Then we have*

$$\sup_{\substack{A \subseteq \mathbb{R}^d \\ A \text{ convex}}} |\mathbb{P}(S_n \in A) - \mathbb{P}(Y \in A)| \leq \frac{400 d^{1/4} \beta}{\sqrt{n}}$$

We need a mildly modified version of this theorem to deal with iid random vectors $X_1, \ldots, X_n$ with non-identity covariance matrices. To this end, suppose that $\Sigma$ is the (full-rank) covariance matrix of each $X_i$, with decomposition $\Sigma = RR^\top$, for some invertible matrix $R$; $R$ can be obtained, for example, by using Cholesky or Schur decomposition. The random variables $R^{-1}X_1, \ldots, R^{-1}X_n$ are then iid, mean zero and with identity covariance matrices, so we may apply Theorem 2 to obtain:

$$\sup_{\substack{A \subseteq \mathbb{R}^d \\ A \text{ convex}}} |\mathbb{P}(R^{-1}S_n \in A) - \mathbb{P}(Y \in A)| \leq \frac{400 d^{1/4} \beta}{\sqrt{n}} \, ,$$

where $\beta = \mathbb{E}\left[\|R^{-1}X_i\|^3\right]$. Now note that this is equivalent to

$$\sup_{\substack{A \subseteq \mathbb{R}^d \\ A \text{ convex}}} |\mathbb{P}(S_n \in RA) - \mathbb{P}(RY \in RA)| \leq \frac{400 d^{1/4} \beta}{\sqrt{n}} \, ,$$

noting that $RY \sim N(0, \Sigma)$.

Since $R$ is invertible, and recalling the definition of the distance $d$ above, this is exactly equivalent to:

$$d(S_n, RY) \leq \frac{400 d^{1/4} \beta}{\sqrt{n}} \, , \tag{16}$$

which is the variant of Bentkus' result we will require in the sequel.

We apply this bound to the sum

$$\sum_{j=1}^{H_{\mu-1}} \widetilde{g}_j^{(\mu-1)}(\mathbf{x}) \otimes w_{\bullet,j}^{(\mu)}$$

Noting that the summands indexed by $j$ are iid by assumption, with the expected third moment norm featuring in the Berry-Esseen inequality upper-bounded by $\beta \leq C_{H_\mu, n}$, for some constant $C_{H_\mu, n}$ depending on $H_\mu$ and $n$, but independent of $H_{\mu-1}$ (finiteness of $C_{H_\mu, n}$ follows from Lemma 4).

As a consequence, we have the following bound:

$$d\left(\sum_{j=1}^{H_{\mu-1}} \widetilde{g}_j^{(\mu-1)}(\mathbf{x}) \otimes w_{\bullet,j}^{(\mu)}, Z'(\mathbf{x})\right) \leq 400 C_{H_\mu, n} (n H_\mu)^{1/4} / \sqrt{H_{\mu-1}} \, ,$$

where $Z'(\mathbf{x}) = (Z_j'(x[i]) | j = 1, \ldots, H_\mu, \ i = 1, \ldots, n)$ is mean-zero multivariate normal, with blockwise independence with respect to the index $j$. We wish to demonstrate that this is less than or equal to $2^{-(D-(\mu-1))-n\sum_{k=\mu}^{D} H_k} \varepsilon$ when $H_D$ is sufficiently large. This is equivalent to showing that

$$400 C_{H_\mu, n} (n H_\mu)^{1/4} 2^{((D+1)-(\mu-1))+n\sum_{k=\mu}^{D} H_k} / \sqrt{H_{\mu-1}} \leq \varepsilon$$

for all sufficiently large $H_D$. But note that with $H_{k-1} = 2^{H_k^2}$ for $k = \mu, \ldots, D-1$, the left-hand side converges to 0 as $H_D$ increases (using the bound obtained for $C_{H_\mu, n}$ in Lemma 4), so for all $H_D$ sufficiently large, we obtain

$$d\left(\sum_{j=1}^{H_{\mu-1}} \widetilde{g}_j^{(\mu-1)}(\mathbf{x}) \otimes w_{\bullet,j}^{(\mu)}, Z'(\mathbf{x})\right) \leq 2^{-((D+1)-(\mu-1))-n\sum_{k=\mu}^{D} H_k} \varepsilon \, ,$$

as required.

Adding the independent bias vector $b^{(\mu)}$ immediately yields

$$d\left(1_n \otimes b^{(\mu)} + \sum_{j=1}^{H_{\mu-1}} \widetilde{g}_j^{(\mu-1)}(\mathbf{x}) \otimes w_{\bullet,j}^{(\mu)}, Z(\mathbf{x})\right) \leq 2^{-((D+1)-(\mu-1))-n\sum_{k=\mu}^{D} H_k} \varepsilon \, ,$$

where $Z(\mathbf{x})$ is mean-zero multivariate normal, with the same block-diagonal covariance structure as described for $Z'(\mathbf{x})$ above, and $1_n \in \mathbb{R}^n$ is a vector of 1's. □

*Proof of Lemma 3.* Let $A \subseteq \mathbb{R}^{nH_\mu}$ be an arbitrary convex set. First, observe that we have

$$
\mathbb{P}\left(\left(1_n \otimes b^{(\mu)} + \sum_{j=1}^{H_{\mu-1}} \phi(\widetilde{f}_j^{(\mu-1)}(\mathbf{x})) \otimes w_{\bullet,j}^{(\mu)}\right) \in A\right)
$$

$$
= \mathbb{E}\left[\mathbb{P}\left(\left(1_n \otimes b^{(\mu)} + \sum_{j=1}^{H_{\mu-1}} \phi(\widetilde{f}_j^{(\mu-1)}(\mathbf{x})) \otimes w_{\bullet,j}^{(\mu)}\right) \in A \middle| w^{(\mu)}, b^{(\mu)}\right)\right]
$$

$$
= \mathbb{E}\left[\mathbb{P}\left(\left(\sum_{j=1}^{H_{\mu-1}} \phi(\widetilde{f}_j^{(\mu-1)}(\mathbf{x})) \otimes w_{\bullet,j}^{(\mu)}\right) \in A - 1_n \otimes b^{(\mu)} \middle| w^{(\mu)}, b^{(\mu)}\right)\right] \qquad (17)
$$

Now, note that for fixed $w^{(\mu)}$ and $b^{(\mu)}$, the event

$$
\left\{\left(\sum_{j=1}^{H_{\mu-1}} \phi(\widetilde{f}_j^{(\mu-1)}(\mathbf{x})) \otimes w_{\bullet,j}^{(\mu)}\right) \in A - 1_n \otimes b^{(\mu)} \middle| w^{(\mu)}, b^{(\mu)}\right\}
$$

is exactly that the vector $\phi(\widetilde{f}^{(\mu-1)}(\mathbf{x}))$ lies in the preimage of the convex set $A - 1_n \otimes b^{(\mu)}$ under the linear map $w^{(\mu)}$, which is again a convex set. Secondly, observe that for the specific ReLU nonlinearity $\phi$, if $C$ is an arbitrary convex set, then $\{(f^{(\mu-1)}(\mathbf{x})|\phi(f^{(\mu-1)}(\mathbf{x})) \in C\}$ may be written as the disjoint union of at most $2^{nH_{\mu-1}}$ convex sets:

$$
\{f^{(\mu-1)}(\mathbf{x})|\phi(f^{(\mu-1)}(\mathbf{x})) \in C\}
$$
$$
= (\phi^{-1}(C) \cap \{t \in \mathbb{R}^{nH_{\mu-1}}|t_i \geq 0 \,\forall i\}) \cup
$$
$$
\bigcup_{\substack{I \subseteq \{1,\ldots,nH_{\mu-1}\} \\ I \neq \emptyset}} \{t \in \mathbb{R}^{nH_{\mu-1}}|t_I < 0, \exists y \in C \text{ s.t. } y_{I^c} = t_{I^c}, y_I = 0\} .
$$

Applying the assumed bound in the statement of the lemma to each of these sets, we obtain

$$
|\mathbb{P}(\phi(f^{(\mu-1)}(\mathbf{x})) \in C) - \mathbb{P}(\phi(\widetilde{f}^{(\mu-1)}(\mathbf{x})) \in C)| \leq 2^{nH_{\mu-1}}\varepsilon .
$$

Substituting this bound into the conditional probability (17) yields

$$
|\mathbb{P}(f^{(\mu)}(\mathbf{x}) \in A) - \mathbb{P}(\widetilde{f}^{(\mu)}(\mathbf{x}) \in A)| \leq 2^{nH_{\mu-1}}\varepsilon .
$$

Since $A$ was an arbitrary convex set, the proof is complete. $\qquad \square$

*Proof of Lemma 4.* Note that by independence of $\widetilde{g}_j^{(\mu-1)}(\mathbf{x})$ from $\widetilde{w}_{\bullet,j}^{(\mu)}$ we have that each $X_j$ has mean zero and covariance $\Sigma_\otimes = \Sigma \otimes \hat{C}_w^{(\mu)}I$ where $\Sigma$ is the covariance matrix of $\widetilde{g}_j^{(\mu-1)}(\mathbf{x})$. By standard properties of the Kronecker product, the Schur decomposition of $\Sigma_\otimes$ is $(Q\Lambda Q^T) \otimes (\hat{C}_w^{(\mu)}I)$ where $Q\Lambda Q^T$ is the Schur decomposition of $\Sigma$. Simple algebraic manipulation yields:

$$
\mathbb{E}\left[\|Q_\otimes \Lambda_\otimes^{-1/2} Q_\otimes^T (\widetilde{g}_j^{(\mu-1)}(\mathbf{x}) \otimes \widetilde{w}_{\bullet,j}^{(\mu)})\|^3\right] = \mathbb{E}\left[\|(Q\Lambda^{-1/2}Q^T\widetilde{g}_j^{(\mu-1)}(\mathbf{x})) \otimes ((\hat{C}_w^{(\mu)})^{-1/2}\widetilde{w}_{\bullet,j}^{(\mu)})\|^3\right]
$$
$$
= \mathbb{E}\left[\|Q\Lambda^{-1/2}Q^T\widetilde{g}_j^{(\mu-1)}(\mathbf{x})\|^3\right]\mathbb{E}\left[\|(\hat{C}_w^{(\mu)})^{-1/2}\widetilde{w}_{\bullet,j}^{(\mu)}\|^3\right] .
$$

Notice that the random variable $(\hat{C}_w^{(\mu)})^{-1/2}\widetilde{w}_{\bullet,j}^{(\mu)}$ follows the $\mathbb{R}^{H_\mu}$-dimensional standard normal distribution, and thus its squared norm follows the chi-squared distribution with $H_\mu$ degrees of freedom, which is also known as the Gamma$(H_\mu/2, 1/2)$ distribution. Exponentiating to the power of $3/2$ and taking the expectation, we obtain:

$$
\mathbb{E}\left[\|(\hat{C}_w^{(\mu)})^{-1/2}\widetilde{w}_{\bullet,j}^{(\mu)}\|^3\right] = 2^{3/2}\frac{\Gamma((H_\mu+3)/2)}{\Gamma(H_\mu/2)} .
$$

Finally, $\|Q\Lambda^{-1/2}Q^T\widetilde{g}_j^{(\mu-1)}(\mathbf{x})\|^3 \leq \|\widetilde{g}_j^{(\mu-1)}(\mathbf{x})\|^3/\lambda_{\min}^{3/2}$ where $\lambda_{\min}$ is the smallest value on the diagonal of $\Lambda$. If the activation $\phi$ does not increase the norm of the input vector (as is the case for rectified linear), we have $\|\widetilde{g}_j^{(\mu-1)}(\mathbf{x})\|^3 \leq \|Z_j^{(\mu-1)}(\mathbf{x})\|^3$ almost surely, where $Z_j^{(\mu-1)}(\mathbf{x})$ follows the known n-dimensional normal distribution with mean zero and covariance matrix whose Schur decomposition will be denoted as $U\Psi U^T$. Using standard Gaussian identities, we can write

$$\mathbb{E}\left[\|Z_j^{(\mu-1)}(\mathbf{x})\|^3\right] = \mathbb{E}\left[\|U\Psi^{1/2}\varepsilon\|^3\right] = \mathbb{E}\left[\|\Psi^{1/2}\varepsilon\|^3\right] \leq 2^{3/2}\psi_{\max}^{3/2}\frac{\Gamma((n+3)/2)}{\Gamma(n/2)},$$

where $\varepsilon \sim \mathcal{N}(0, I_n)$ and $\psi_{\max}$ is the highest entry on the diagonal of $\Psi$. Putting it all together, we arrive at the desired upper bound $C_{H_\mu,n}$

$$\mathbb{E}\left[\|Q_\otimes\Lambda_\otimes^{-1/2}Q_\otimes^T X_j\|^3\right] \leq \left(4\frac{\psi_{\max}}{\lambda_{\min}}\right)^{3/2}\frac{\Gamma((H_\mu+3)/2)}{\Gamma(H_\mu/2)}\frac{\Gamma((n+3)/2)}{\Gamma(n/2)}.$$

Because $\psi_{\max}$ and $\lambda_{\min}$ are derived from the distribution of the limiting variable $\widetilde{g}_j^{(\mu-1)}(\mathbf{x}) = \phi(Z_j^{(\mu-1)}(\mathbf{x}))$, which only depends on $\mu$, the bound only depends on $H_\mu$ and $n$ as desired. Further, noting that $\Gamma((x+3)/2)/\Gamma(x/2) = \mathcal{O}(x^2)$, we have that $C_{H_\mu,n} = \mathcal{O}(H_\mu^2 n^2)$, as required. $\quad\square$

*Proof of Proposition 1.* We first apply Lemma 3 to the assumed inequality

$$d(f^{(\mu-1)}(\mathbf{x}), Z^{(\mu-1)}(\mathbf{x})) \leq 2^{-((D+1)-(\mu-1))-n\sum_{k=\mu-1}^{D}H_k}\varepsilon,$$

to obtain

$$d(f^{(\mu)}(\mathbf{x}), \widetilde{f}^{(\mu)}(\mathbf{x})) \leq 2^{-((D+1)-(\mu-1))-n\sum_{k=\mu}^{D}H_k}\varepsilon.$$

We apply Lemma 2 so that for $H_D$ sufficiently large, we have

$$d(\widetilde{f}^{(\mu)}(\mathbf{x}), Z^{(\mu)}(\mathbf{x})) \leq 2^{-((D+1)-(\mu-1))-n\sum_{k=\mu}^{D}H_k}\varepsilon.$$

Applying the triangle inequality then yields

$$d(f^{(\mu)}(\mathbf{x}), Z^{(\mu)}(\mathbf{x})) \leq 2^{-((D+1)-\mu)-n\sum_{k=\mu}^{D}H_k}\varepsilon,$$

as required. $\quad\square$

*Proof of Proposition 2.* The idea of the proof is to chain Proposition 1 together across the layers of the network. We fix $\varepsilon > 0$, and apply Proposition 1 to each layer of the network, yielding the following set of implications for $H_D$ sufficiently large:

$$d(f^{(\mu-1)}(\mathbf{x}), Z^{(\mu-1)}(\mathbf{x})) \leq 2^{-((D+1)-(\mu-1))-n\sum_{k=\mu-1}^{D}H_k}\varepsilon$$

$$\implies d(f^{(\mu)}(\mathbf{x}), Z^{(\mu)}(\mathbf{x})) \leq 2^{-((D+1)-\mu)-n\sum_{k=\mu}^{D}H_k}\varepsilon,$$

for $\mu \in \{2, \ldots, D+1\}$. Finally, note that from the definition of the network, the distribution of $f^{(1)}(\mathbf{x})$ is exactly multivariate normal with the required covariance structure, so that $d(f^{(1)}(\mathbf{x}), Z^{(1)}(\mathbf{x})) = 0$, completing the proof. $\quad\square$

*Proof of Theorem 1.* To prove that $(f^{(D+1)}(x[i]))_{i=1}^{\infty}$ converges weakly to a Gaussian process with respect to the metric $\rho$ on $\mathbb{R}^{\mathbb{N}}$ given by:

$$\rho(v, v') = \sum_{i=1}^{\infty} 2^{-i}\min(1, |v_i - v_i'|) \qquad \forall v, v' \in \mathbb{R}^{\mathbb{N}},$$

it is sufficient (Billingsley, 1999, p. 19) to prove weak convergence of the finite-dimensional marginals of the process to multivariate Gaussian random variables, with covariance matrix matching that specified by the kernel of the proposed Gaussian process.

To this end, let $I$ be a finite subset of $\mathbb{N}$, and consider the inputs $(x[i])_{i\in I}$. We may now apply Proposition 2 to obtain weak convergence of the joint distribution of the output variables of the network, $(f^{(D+1)}(x[i]))_{i\in I}$ to a multivariate Gaussian with the correct covariance matrix. As the finite subset of inputs was arbitrary, we are done. $\quad\square$

