# OpenReview forum: "Gaussian Process Behaviour in Wide Deep Neural Networks"
_ICLR.cc/2018/Conference — Accept (Poster)_

### Official Review · AnonReviewer1 · 2017-11-27
**A nicely written paper for the most part, but practical value is unclear**

**Rating:** 6
**Confidence:** 4

**Review:**

The authors study the limiting behaviour for wide Bayesian neural networks, comparing to Gaussian processes.

The paper is well written, and the experiments are enlightening. This work is a nice follow up to Neal (1994), and recent work considering similar results for neural networks with more than one hidden layer. It does add to our understanding of this body of work.

The weakness of this paper is in its significance and practical value. This infinite limit loses much of the interesting representation in neural networks because the variance of the weights goes to zero. Thus it’s unclear whether these formulations will have many of the benefits of standard neural networks, and whether they’re particularly related to standard neural networks at all. There also don’t seem to be many practical takeaways from the experiments, and the experiments themselves do not consider any predictive tasks at all. It would be nice to see some practical benefit for a predictive task actually demonstrated in the paper. I am not sure what exactly I would do differently in training large neural networks based on the results of this paper, and the possible takeaways are not tested here on real applications.

This paper also seems to erroneously attribute this limitation of the Neal (1994) limit, and its multilayer extensions, to Gaussian processes in the section “avoiding Gaussian process behaviour”. The problems with that construction are not a profound limitation of Gaussian processes in general. If we can learn the kernel function, then we can learn an interesting representation that does not have these limitations and still use a GP. We could alternatively treat the kernel parameters probabilistically, but the fact that in this case we would not marginally have a GP any longer is mostly incidental. The discussed limitations are more about specific kernel choices, and lack of kernel learning, than about “GP behaviour”.

Indeed, while the discussion of related work is mostly commendable, the authors should also discuss the recent papers on “deep kernel learning”:
i) http://proceedings.mlr.press/v51/wilson16.pdf
ii) https://papers.nips.cc/paper/6426-stochastic-variational-deep-kernel-learning.pdf
iii) http://www.jmlr.org/papers/volume18/16-498/16-498.pdf

In particular, these papers do indeed learn flexible representations with Gaussian processes by using kernels constructed with neural networks. They avoid the behaviour discussed in the last section of your paper, but still use a Gaussian process. The network structures themselves are trained through the marginal likelihood of the Gaussian process. This approach effectively learns an infinite number of adaptive basis functions, parametrized through the structural properties of a neural network. Computations are made scalable and practical through exploiting algebraic structure.

Overall I enjoyed reading your paper.

---

> ### Author Response · Authors · 2017-12-21
> **AnonReviewer1 author response**
>
> Thank you for your detailed and thought provoking review.  We will acknowledge your anonymous contribution in the final version of the paper.
>
> --On the deep kernel learning papers of Wilson et al and Al-Shedivat et al:
>
> We agree that this deep kernel literature is useful and relevant in this context. We are sure you would agree that is not the only promising approach. In Section 6 we did point out that the “emergent kernels in our case are hyperparameter free” and that “any Gaussian process with a fixed kernel does not use a learnt hierarchical representation”. Therefore we respectfully disagree with your assessment that we “erroneously attributed” this behaviour to GP methods with a learnt kernel. Nevertheless, we agree that the paper would be clearer with more discussion of learnt representations and we have added additional material to Section 6 along with the citations you kindly suggested.
>
> --On significance and practical value:
>
> We agree that, in your words: “this infinite limit loses much of the interesting representation in neural networks because the variance of the weights goes to zero.” Indeed, the careful extension of the mathematics underlying this intuition to networks with more than one hidden layer is part of the contribution of our paper. We view the cautionary message of the paper as one of its key scientific contributions. Furthermore, Neal's original 1996 work "suffers" from the same issue yet has become extremely influential and led to many invaluable insights. Our analysis moves the careful study of random networks beyond what was known. This requires considerable technical insight. The theoretical assumptions we make are less restrictive than for instance Daniely et al. (2016), which was (correctly in our opinion) regarded as impactful at that NIPS.
>
> Although, as we acknowledge, it is difficult to do exhaustive experiments in the fully Bayesian regime, our experiments with the base network architecture of Hernandez-Lobato and Adams (2015) suggest that the Gaussian process limit is relevant to wide finite Bayesian neural networks in the regime studied.

---

### Official Review · AnonReviewer3 · 2017-11-28
**Adds to the theoretical understanding of the deep wide regime in NNs, no clear application**

**Rating:** 6
**Confidence:** 4

**Review:**

- Summary

The paper is well written and proves how deep, wide, fully connected NNs are equivalent to GPs in the limit. This result, which was well known for single-layer NNs, is now extended to the multilayer case. Although there was already previous work suggesting GP this behavior, there was no formal proof under the specific conditions presented here.

The convergence to a GP is also verified experimentally on some toy examples.


- Relevance

The result itself does not feel very novel because variants of it were already available.

Unfortunately, although making other researchers aware of this is worthy, the application of this result seems limited, since in fact it describes and lets us know more about a regime that we would rather avoid, rather than one we want to exploit. Most of the applications of deep learning benefit from strong structured priors that cannot be represented as a GP. This is properly acknowledged in the paper.

The lack of practical relevance combined with the not-groundbreaking novelty of the result makes this paper less appealing.


- Other comments

Page 6: "It does mean however that our empirical study does not extend to larger datasets where such inference is prohibitively expensive (...) prior dominated problems are generally regarded as an area of strength for Bayesian approaches and in this context our results are directly relevant."

Although that argument can hold for datasets that are large in terms of amount of data points, it doesn't for datasets that are large in terms of number of dimensions. The empirical study could have used very high-dimensional datasets with comparatively low amounts of training data. That would maintain a regime were the prior does matter but and better show the generality of the results.

Page 6: "We use rectified linear units and correct the variances to avoid a loss of prior variance as depth is increased as discussed in Section 3"

Are you sure this is discussed in Section 3?

Page 4: "This is because for finite H the input activations do not have a multivariate normal distribution".

Can you elaborate on this? Since we are interested in the infinite limit, why is this a problem?

---

> ### Author Response · Authors · 2017-12-21
> **AnonReviewer3 author response**
>
> Thank you for your review which raises some important questions. We will endeavour here to answer them more clearly. We will acknowledge your anonymous contribution in the final version of the paper.
>
> If you will excuse us we will start with your last question first since it relates to your criticism of the significance of the work.
>
> From your review: Page 4: ` "This is because for finite H the input activations do not have a multivariate normal distribution".  Can you elaborate on this? Since we are interested in the infinite limit, why is this a problem?'
>
> This is an important point. There is a general answer and a more specific answer:
>     1) In general, weak convergence is exactly that - many general manipulations that we might want to perform with it don't actually hold. For instance if the sequence of distributions ( a_n ) converges weakly/in-distribution to a and the sequence of distributions ( b_n ) converges weakly/in-distribution to b then the sequence of independent product distributions (a_n,b_n) doesn't necessarily converge weakly/in-distribution to (a,b). See Billingsley 1999 page 23. Care and rigour is required in this domain.
>     2) More specifically to this example, the rate at which the convergence of the activations occurs could have a ``knock on effect'' on the convergence of the activation distributions further through the network. We've added a comment about this second point to the main text just after the sentence you quote.
>
> As an example of point 2). Suppose that the sequence of distributions (P_n) converges in distribution to some P_*. Consider the limit of a sequence of expectations (\int \psi_n d P_n ) where the integrand is also changing. This will not in general be the same as if we first substitute the limit measure (\int \psi_n d P_*) and then take the n limit of the new integral. The rate of convergence will in general matter.
>
> From your review: "The result itself does not feel very novel because variants of it were already available."
>
> We have already argued that improving rigorous results in this area is very desirable. Therefore we must respectfully disagree. To the best of our knowledge there are no rigorous results about convergence in this area since Neal (1996).
>
> It is fair to point out that our empirical analysis does not extend to high dimensional functions- thank you. We've updated the discussion to reflect this. Note that the content of Theorem 1 does not depend on the dimensionality of the inputs.
>
> Also from your review: "Are you sure this is discussed in Section 3?"
>
> You are correct - we do not allude to this. Thank you for pointing this out. This is an orphaned cross reference to some material that did not make the cut because it is orthogonal to the main thrust of the paper. Essentially, carefully scaling the weight variances can help mitigate the onset of the depth pathologies discussed in Duvenaud et al (2014). We apologize and have now removed this. The exact code we used is available in our anonymous repository.

---

### Official Review · AnonReviewer2 · 2017-11-29
**The paper focuses on proving and discussing properties of wide deep neural networks, and more particularly of their behaviour when priors on the weights are assumed.**

**Rating:** 6
**Confidence:** 4

**Review:**

In part 1, the authors introduce motivation for studying wide neural networks and summarize related work.
In part 2,  they present a theorem (main theoretical result) stating that under conditions on the weight priors, the output function of a multi-layer neural network (conditionally to a given input) weakly converges to a gaussian process as the size of the hidden layers go to infinity.
remark on theorem 1: This result generalizes a result proven in 2015 stating that the normality of a layer propagates to the next as the size of the first layer goes to infinity. The result stated in this paper is proven by bounding the gap between the output distribution and the corresponding gaussian process, and by propagating this bound across layers (appendix).
In part 3, the authors discuss the choice of a nonlinearity function that enables easy computation of the kernels introduced in the covariance matrix of the limit normal distribution. Their choice lands on ReLU.
In part 4, the focus is on the speed of the convergence presented in theorem 1. Experiments are conducted to show how the distance (maximum mean disrepancy) between the output distribution and its theoretical gaussian process limit vary when the sizes of the hidden layers increase. The results show that the convergence (in MMD) happens consistently, although it is slower when the number of hidden layers gets bigger.
In part 5, the authors compare the distributions (finite Bayesian deep networks and their analogues Gaussian processes) in yet another way: by studying their agreement in terms of inference. For this purpose, the authors chose several crieteria: the first two moments of the posterior, the log marginal likelihood and the predictive log-likelihood. The authors judge that the distributions agree on those criteria, but do not provide further analysis.
In part 6, now that It has been shown that the output distributions of Bayesian neural nets do not only weakly converge to Gaussian processes but also behave similarly in terms of inference, the authors discuss ways to avoid the gaussian process behaviour. Indeed, it seems that Gaussian processes with a fixed kernel cannot learn hierarchical representations, that are essential in deep learning.
The idea to avoid the Gaussian process behaviour is to contradict one of the hypothesis of the CLT (so that it does not hold anymore), either by controlling the size of intermediate layers, by using networks with infinite variance in the activities, or by choosing non-independent weights.
In part 7, it is concluded that the result that has been proven for size of layers going to infinity (Theorem 1) seems to empirically be verified on finite networks similar to those used in the literature. This can be used to simplify inference in cases were the gaussian process behaviour is desired, and opens questions on how to avoid this behaviour the rest of the time.

Pros: The authors line of thought of the authors is overall quite easy to follow. The main theoretical convergence result is stated early on, and the remaining of the article is dedicated to observing this result empirically from different angles (MMD, inference, predictive capability..). The last part contains a discussion concerning the extent to which it is actually a desired or a undesired result in classical deep learning use-cases, and the authors provide intuitive conditions under which the convergence would not hold. The stated theorem is a clear improvement on the past literature and is promising in a context where multi-layers neural networks are more and more studied.
Finally, the work is well documented.

Cons:
I have a some concerns with the main result (Theorem 1) and found that some of the notations / formulas were not very clear.
 Concerns with Theorem 1:
* at the end of the proof of Lemma 2, H_\mu is to be chosen large enough in order to get the \epsilon bound of the statement. However, I think that  H_\mu is constrained by the statement of Proposition 2, not to be larger than a constant times 2^(H_{\mu+1}). Isn't that a problem?
* In the proof of Lemma 4, it looks like matrix \Psi, from the schur decomposition of \tilde f, actually depends on H_{\mu-2}, thus making \psi_max depend on it too, as well as the final \beta bound, which would contradict the statement that it depends only on n and H_{\mu}. Could you please double check?

Unclear statements/notations:
* end of page 3, notations are not entirely consist with previous notations
* I do not understand which distribution is assumed on epsilon and gamma when taking the expectancy in equation (9).
* the notation x^(i) (in the theorem and the proof notably) could be changed, for the ^(i) index refers to the depth of the layer in the rest of the notations, and is here surprisingly referring to a set of observations.
* the statement of Theorem 1:
    * I would change "for a countable input set" to "for any countable input set", if this holds true.
    * does not say that the width has to go to infinity for the convergence to happen, which goes a bit in contradiction with the adjective "wide". However, the authors say that in practice, they use the identity as width function.
* I understood that the conclusion of part 3 was that the expectation of eq (9) was elegantly computable for certain non-linearity (including ReLU). However I don't see the link with the "recursive kernel" idea (maybe it's just the way to do the computation described in Cho&Saul(2009) ?)

Some places where it appears that there are minor mistakes:
* 7th line from the bottom of page 3, the vector f^{(2)}(x) contains f_i^{(1)}(x) but should contain f_i^{(2)}(x)
* last display of page 3: change x and x', and indicate upper limit of the sum
* please double check variances C_w and/or \hat{C}_w appearing in equations in (9) and (13).
* line 2 of second paragraph after equations (8) and (9). The authors refer to equation (8) concerning the independence of the components of the output. I think they rather wanted to refer to (9). Same for first sentence before eq (14).
* middle of page 12: matrix LY should be RY.

---

> ### Author Response · Authors · 2017-12-21
> **AnonReviewer2 author response**
>
> We thank the reviewer for their careful reading of the paper. We will acknowledge your anonymous contribution in the final version of the paper.
>
> Regarding the technical query for the proof of Lemma 2, we now have slightly rearranged the material to make clear what constitutes a “sufficiently large H_\mu for the bound to hold, and to show that this is consistent with the growth rates in the statement of Lemma 2. In fact, we require a rate which grows faster than 2^{n H_\mu}, for all n to deal with this, and so have adjusted the stated growth rates in Lemma 2 to H_{\mu-1} = O(2^{H_{\mu}^2}).
>
> Regarding Lemma 4, the bound is actually independent of H_{\mu-2}. This is because \tilde g^{\mu - 1} is a deterministic transformation of Z^{\mu - 1} with the known n-dimensional normal distribution from Lemma~1, independent of H_{\mu-2}. The original proof incorrectly bounded the norm of \tilde g^{\mu - 1} in terms of \tilde f^{\mu - 1} instead of Z^{\mu - 1} which we noticed thanks to your comment.
>
> These details have now been incorporated into the relevant sections of the appendix. We emphasise that the conclusion of Theorem 1 in the main paper remains unchanged, and thank the reviewer once again for their close attention to the details of the proof.
>
> Regarding notational issues:
> Bottom of p3 - we have modified the notation to be consistent with the rest of the paper.
> Eq (9) - we have now defined the distributions in the text.
> x^{(i)} notation - we now use notation of the form x[i] to refer to the different input points to the neural network.
> Theorem 1 - we have made the suggested change of wording, and added the phrase “strictly increasing” to emphasise that the convergence happens as layer widths go to infinity.
> Recursive kernel comment: Cho & Saul (2009) indeed solve the~integral recursion of Hazan and Jaakkola (2015); we have provided more details to make the link clearer.
>
>
> Minor mistakes:
> Thank you for spotting these, we have made the relevant changes in the revised version of the paper.

---

### Author Response · Authors · 2018-08-15
**Extended version of the paper**

A significantly extended version of this paper can be found at https://arxiv.org/abs/1804.11271 . We proved a more general result and added additional experiments. See section 1.2 of the new version for more discussion of the comparison to the version here on OpenReview that was accepted to ICLR. The authors think the new paper is a significant advance.

---

### Decision · Program_Chairs · 2018-01-29
**ICLR 2018 Conference Acceptance Decision**

**Decision:**

Accept (Poster)

**Comment:**

 A clearly written paper. While the practical relevance that came up in the review remains, the analysis and discussion is important for a deeper understanding of the deeper connections between these two important areas of machine learning.